# Piezo1 forms a slowly-inactivating mechanosensory channel in mouse embryonic stem cells

Josefina Inés del Mármol[1†], Kouki K Touhara[1], Gist Croft[2], Roderick MacKinnon[1]*

[1]Laboratory of Molecular Neurobiology and Biophysics, Rockefeller University, Howard Hughes Medical Institute, New York, United States; [2]Laboratory of Stem Cell Biology and Molecular Embryology, Rockefeller University, New York, United States

**Abstract** Piezo1 is a mechanosensitive (MS) ion channel with characteristic fast-inactivation kinetics. We found a slowly-inactivating MS current in mouse embryonic stem (mES) cells and characterized it throughout their differentiation into motor-neurons to investigate its components. MS currents were large and slowly-inactivating in the stem-cell stage, and became smaller and faster-inactivating throughout the differentiation. We found that Piezo1 is expressed in mES cells, and its knockout abolishes MS currents, indicating that the slowly-inactivating current in mES cells is carried by Piezo1. To further investigate its slow inactivation in these cells, we cloned Piezo1 cDNA from mES cells and found that it displays fast-inactivation kinetics in heterologous expression, indicating that sources of modulation other than the aminoacid sequence determine its slow kinetics in mES cells. Finally, we report that Piezo1 knockout ES cells showed a reduced rate of proliferation but no significant differences in other markers of pluripotency and differentiation.
DOI: https://doi.org/10.7554/eLife.33149.001

*For correspondence:
mackinn@mail.rockefeller.edu

Present address: †Laboratory of Neurophysiology and Behavior, Rockefeller University, New York, United States

Competing interests: The authors declare that no competing interests exist.

## Introduction

Elucidating the molecular mechanisms of mechanosensation stands as one of the most exciting challenges in sensory biology. The complexity of the physiological stimuli involved (e.g. osmotic changes, shear stress, cell contact) and the broad timescales in which they operate suggest the existence of equally complex underlying transduction machineries. The recent identification of a structurally unique family of mechanosensory ion channels, the Piezos (*Coste et al., 2010*), and the plethora of mechanosensory processes to which they have since been associated, certainly re-kindled the enthusiasm. Piezos are present in most eukaryotes (with the notable exception of yeasts) (*Xiao and Xu, 2010*) where they mediate light-touch sensation (*Ranade et al., 2014b*), vascular endothelial development (*Li et al., 2014*), and respiratory control (*Nonomura et al., 2017*), to name just a few. Though modest variations have been reported (*Schneider et al., 2017*; *Woo et al., 2014*; *Ikeda et al., 2014*), in their canonical forms, they inactivate quickly following stimulation with a time constant less than 20 ms and in a voltage-dependent manner (*Coste et al., 2010*; *2012*). In mammals, loss of function by knockout of either member of the Piezo family is either unviable (*Ranade et al., 2014b*; *2014a*), or severely debilitating (*Chesler et al., 2016*; *Delle Vedove et al., 2016*; *Haliloglu et al., 2017*) but gain of function mutations can also result in severe defects: mutations in the human *PIEZO1* gene that cause slow inactivation have recently been associated with hereditary xerocytosis, a disorder of ionic imbalance in red blood cells (*Albuisson et al., 2013*; *Bae et al., 2013*). These discoveries highlight the importance of a tight regulation in expression and kinetics of mechanosensory ion channels.

Notably, multiple cell lines exhibit a variety of undescribed stretch-activated currents that differ from Piezos in their kinetics. For example, dorsal root ganglia cells display three types of mechanosensory ionic currents when directly stimulated with a probe: rapid-, intermediate-, and slow-inactivating currents (*Coste et al., 2010*). Piezo2 only accounts for the rapid-inactivating responses, with slow- and non-inactivating conductances still uncharacterized. Other cultured cell lines like C2C12 also express a form of slow-inactivating mechanosensory current, also not yet characterized (*Coste et al., 2010*). Understanding the components of slow-inactivating mechanosensory responses would not only help complete the landscape of mechanosensory ion channels and molecules, but also provide insight into the cellular fine-tuning of responses to diverse stimuli.

We found a large mechanosensitive current in mouse embryonic stem cells with distinctively slow-inactivating kinetics that resembles currents present in C2C12 cells and slow-inactivating DRGs. In addition to a self-standing interest in identifying slow-inactivating mechanosensory components, we found its presence in stem cells particularly interesting. Although not part of a mechanosensory organ, stem cells are extremely alert to environmental cues. Multiple reports show that the cellular fate of multipotent stem cells can be influenced by mechanical strain, shear stress, substrate stiffness or elasticity (*Blumenthal et al., 2014*; *Engler et al., 2006*; *Ivanovska et al., 2015*; *Lu et al., 2016*; *Pathak et al., 2014*). Given the magnitude of these effects, increasing efforts are now focused on elucidating the molecular details of the transduction process.

We describe in this manuscript a large mechanosensitive, slowly-inactivating current in mouse embryonic stem cells. We investigated the evolution of this stem cell mechanosensory current along a model differentiation pathway into motor neurons, and found it to be carried by Piezo1.

## Results

### Mouse embryonic stem cells exhibit a slowly-inactivating mechanosensitive current

We screened multiple cell lines searching for slow inactivating mechanosensitive (MS) currents using the 'poking' assay (*Coste et al., 2010*). In this assay individual cells can be mechanically stimulated with a round-end probe controlled by a piezo-actuator, while a second probe located at a distant part of the cell performs patch-clamp recordings. Mouse embryonic stem cells (mES cells) exhibited robust, slow inactivating MS currents (*Figure 1A*). Currents ranged from 0 to over 2100 pA over baseline, with an average value of 465 ± 112 pA (n = 30). MS currents could not be reliably fit to mono- or bi- exponential functions due to the large variability of the initial decay step. In order to quantify the inactivation behavior we used as a metric the slow inactivating component (slow fraction), defined as the relative fraction of peak current at the beginning of the stimulus that still remained 75 ms into the poking step. For a canonical fast-inactivating channel such as Piezo1 the slow fraction is typically less than 0.2. In mES cells the slow fraction of MS current had an average value 0.67 ± 0.04 (n = 30) and in some cells it approached 1.0.

A detailed study of the MS currents in mES cells revealed that they are selective for cations with a permeability sequence as defined by the reversal potential under bi-ionic conditions: $P_{Ca} > P_K = P_{Na} > P_{NMDG}$ (*Figure 1B*) (*Hille, 1992*, pg 19). In excised membrane patches, stimulation by pressure clamp elicited single mechanosensitive channels that correlated temporally with stimulation (*Figure 2* label panels A, D). An ensemble of multiple pressure-evoked recordings of single channels at the same voltage yielded macroscopic currents whose kinetics mimic the whole cell MS currents activated by poking (*Figure 2D*). Amplitude histograms at multiple voltages revealed a single channel conductance of 24.7 ± 2.5 pS in standard physiological conditions (*Figure 2B and C*).

### MS current in mES cells depends on the differentiation state of the cell

To study the evolution of the MS current after exiting the stem cell state we differentiated mES cells into motor neurons (*Wichterle and Peljto, 2008*). To initiate the differentiation, growth factors are removed from the media to which mouse embryonic stem cells are exposed, which terminates their pluripotent stage and sends them to a state of responsiveness to patterning signals. Retinoic acid (RA) is then applied, which induces differentiation into spinal nerve cells. Further addition of Smoothened agonist (SAG) at day three controls ventralization of nascent spinal neurons. Finally, addition of glial cell derived neurotrophic factor (GDNF) turns on a host of motor neuron-specific genes. To

monitor the differentiation we used Hb9-GFP stem cells, in which the motor neuron specific pro-moter Hb9 drives GFP expression (*Wichterle et al., 2002*). We achieved approximately 30% efficiency as assessed by GFP expression and morphological and functional characterization. Three independent differentiations were performed, the same studies and measurements were carried out each time, and data from all three were combined.

Evolution of voltage-dependent and MS currents were monitored throughout the course of differentiation (*Figure 3B,C*). Voltage-dependent currents in mES cells were small and remained so throughout the first 4 days until neuronal stages were approached. Around day five voltage-dependent potassium currents became larger and some cells began to exhibit small voltage-dependent sodium currents (*Figure 3B*). This, along with low GFP expression, indicated the presence of immature motor neurons. By day 7, GFP expression increased and GFP-positive cells acquired a typical neuronal profile of large voltage-dependent $K^+$ and $Na^+$ currents and were able to fire action potentials. Morphologically the differentiation progressed in a similar manner (*Figure 3A*), with a rather non-differentiated appearance in the initial steps and the presence of neuronal processes as the cells approached days 5 to 7.

MS currents followed a different course of development compared to voltage-dependent currents (*Figure 3C*). MS currents were large and slow-inactivating at the stem cell stage and became increasingly smaller with a diminishing slow fraction as differentiation proceeded. No mechanically-evoked currents were present by day 7. A quantification of the MS currents is summarized in *Figure 4*. *Figure 4A* shows the amplitudes of both the peak and slow-inactivating (measured 75 ms into the poking step) currents throughout the differentiation. *Figure 4B* shows the slow fraction throughout the differentiation. The large spread of data points reflects a high degree of heterogeneity among cells, however, the overall trend is clear.

## Piezo1 forms the slowly inactivating current of mES cells

To study the components of this current we performed expression analysis throughout the differentiation. We found that the expression of Piezo1 correlates with the observed MS current (*Figure 5A, B*). More specifically, expression of the *Piezo1* gene follows the same course as the fast component of the MS current in mES cells. We then used Crispr/Cas9 technology to knock out the *Piezo1* gene from mES cells and identify its contribution to the MS current. To minimize the chance of confounding factors from off-target effects, we obtained two independent clones using two different sgRNA sequences to guide Cas9 nuclease (*Figure 5C*). We obtained two separate colonies of mES cells in which the *Piezo1* gene was knocked out by generating a frameshift mutation, which introduced an early stop codon. Study of the mechanosensitive behavior of these cells revealed that the entirety of the MS current is absent in knockout cells, rendering Piezo1 the likely pore-forming subunit of the MS channel in mES cells (*Figure 5D*).

## Heterologous expression of Piezo1 cDNA from mouse embryonic stem cells yields a fast inactivating MS current

Recent work reported that point mutations of the human *PIEZO1* gene give rise to a version of the protein with slowed inactivation kinetics (*Albuisson et al., 2013*; *Bae et al., 2013*). To investigate whether the conspicuous kinetics of Piezo1 in mES cells is due to an intrinsic feature of the *Piezo1* gene in these cells, we studied the sequence of Piezo1 cDNA from mES cells and compared it to the reference sequence for Piezo1 obtained by the Patapoutian lab from N2A cells (*Coste et al., 2010*). The cDNA recovered from mES cells encodes a protein that contains three amino acid substitutions compared to the original sequence cloned from N2A cells: G147R, I229V, and V1572M. To analyze whether these mutations generate an intrinsically slower channel, we expressed the construct in HEK 293 cells in which we knocked-out of the endogenous *Piezo1* gene to achieve a cell line with a clean background and compared its inactivation kinetics to the canonical Piezo1 cloned by the Patapoutian lab. Both constructs yielded currents with similar, fast inactivation kinetics (*Figure 6A,B*), excluding the point substitutions as accounting for the altered kinetics of Piezo1 in mES cells.

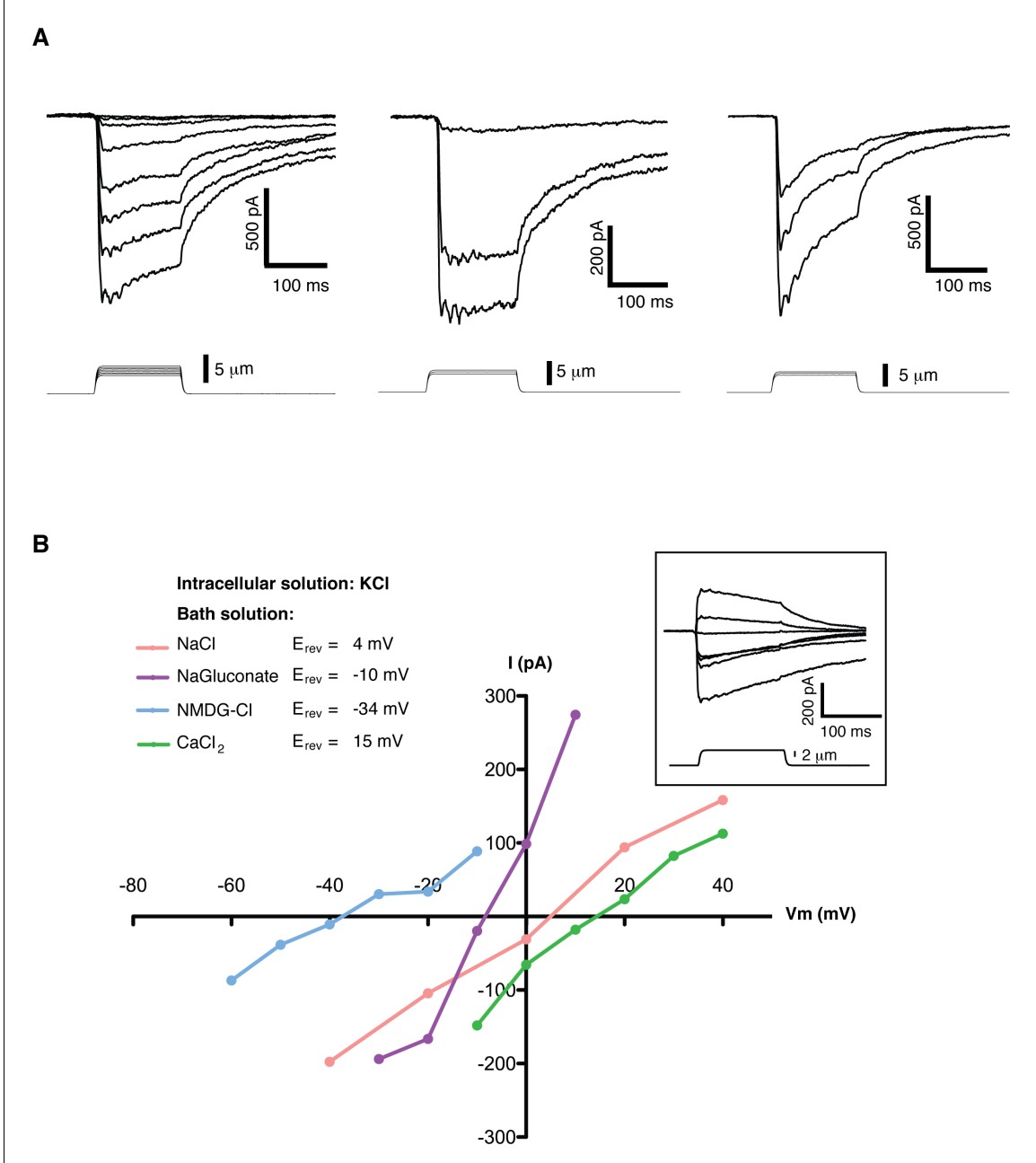

**Figure 1.** Mechanosensitive currents in mouse embryonic stem cells. (**A**) Ionic currents recorded from mouse embryonic stem cells in response to mechanical stimulation. Cells were clamped at −80 mV in whole-cell mode and mechanosensitive currents were elicited by poking steps of increasing depth. Examples from three different cells are shown. (**B**) Mechanosensitive currents were recorded at different voltages under diverse bi-ionic conditions (inset) to establish ion selectivity. Briefly, the current's reversal potential ($E_{rev}$) will shift towards that of its permeant ion for each particular condition. The internal solution always contains KCl. The purple trace corresponds to NaGluconate in the bath, to analyze chloride permeation. If chloride were to permeate the pore, the $E_{rev}$ should move towards that of chloride, infinitely positive in these conditions. The fact that the $E_{rev}$ did not move in that direction at all indicates that chloride does not permeate the pore (the slight shift to negative values can be accounted for by the difference in motility of gluconate, a much slower ion than chloride). The pink curve is done using NaCl in the bath solution. The $E_{rev}$ sits at 4 mV, roughly in between the $E_{rev}$ for potassium and sodium in these conditions, indicating that both ions are equally likely to travel through the pore. The blue curve is in the presence of NMDG in the bath. The $E_{rev}$ moves towards that of potassium, indicating very low permeability for NMDG. Finally, the green curve taken with calcium as the only cation in the bath shifts the $E_{rev}$ to +15 mV, closer to the $E_{rev}$ of calcium than to that of potassium in these conditions, indicating a slightly higher permeability for calcium than for potassium. The observed permeability sequence is then: $P_{Ca} > P_K = P_{Na} > P_{NMDG}$. Raw traces are shown with no voltage corrections. Liquid junction potentials are not corrected, they were estimated as: 4.3 mV for NaCl/KCl, 8.2 mV for $CaCl_2$/KCl, −6.7 mV for NaGluconate/KCl, and 9.3 mV for NMDG-Cl/KCl.

DOI: https://doi.org/10.7554/eLife.33149.002

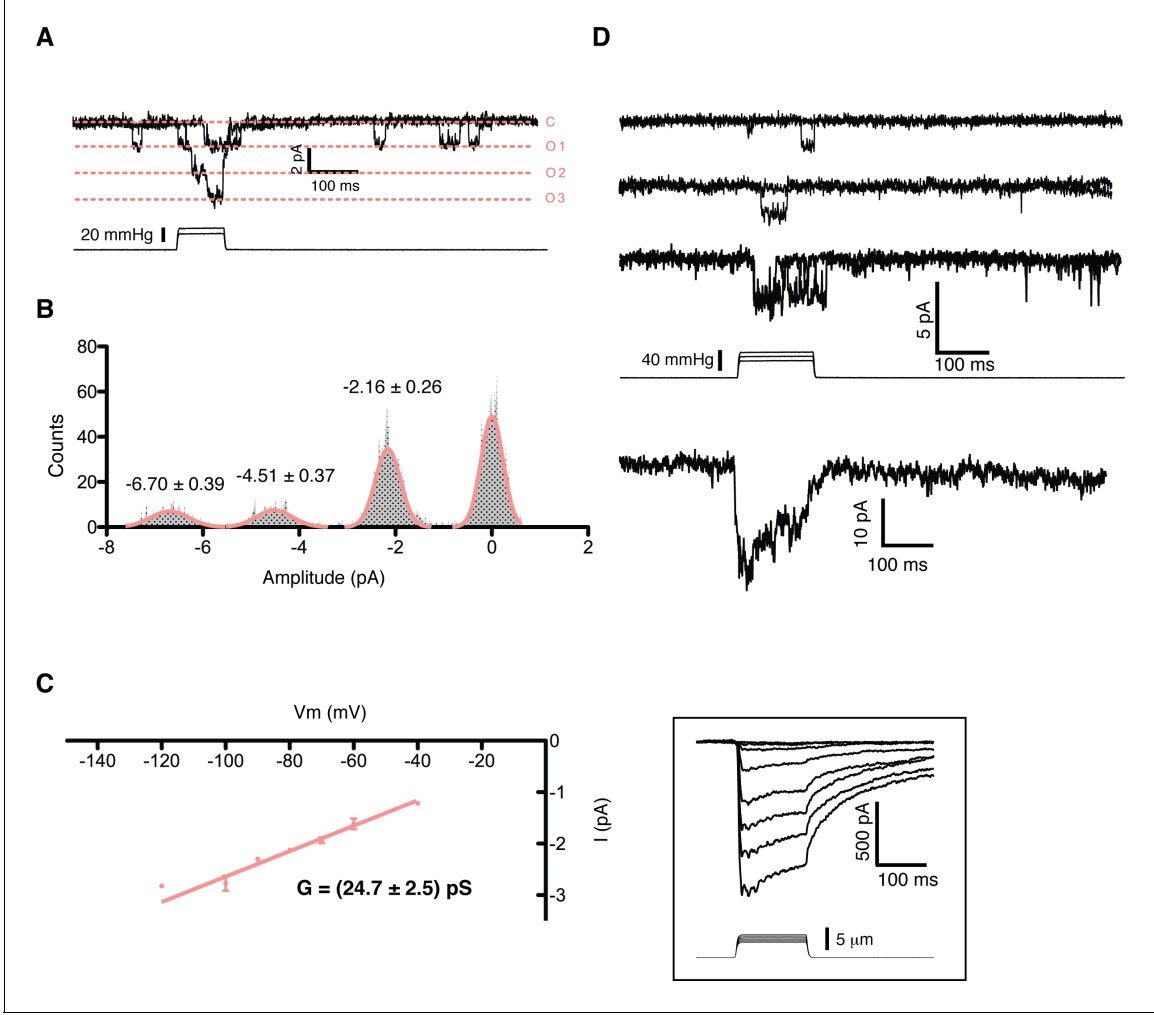

**Figure 2.** Single channel analysis of the mechanosensitive channel in mouse emrbyonic stem cells in outside-out membrane patches. (A) A membrane patch is clamped at −80 mV and channel activity is evoked by pressure steps; the opening of 3 channels can be seen. (B) The amplitude histogram of the recording shown in (A) shows the distribution of all observed states (closed, open1, open2, open3). (C) Single channel conductance estimation from amplitude histograms of multiple recordings at different voltages gives a value of 24.7 ± 2.5 pS. (D) The ensemble of multiple pressure-evoked single channel recordings at −80 mV gives an apparent macroscopic current whose kinetics resembles that of the whole-cell poking currents (whole-cell poking current is shown in an inset for comparison).

DOI: https://doi.org/10.7554/eLife.33149.003

## Piezo1 knockout does not affect pluripotency, early differentiation, or substrate stiffness-response, but decreases proliferation

We next sought to determine the developmental role of Piezo1 by comparing Piezo1 knockout to wild type lines across a series of in vitro pluripotency and early-differentiation phenotypes. During 'naïve' inner cell mass pluripotency (mESC) there was no significant difference in the organization of mESC colonies, expression of pluripotency transcription factors (SOX2), typical cell-to-cell or cell-to-substrate adhesions (E-Cadherin, F-Actin), or size of colonies (*Figure 7A*, *Table 1*). Overall proliferation rates however, were significantly lower in KO compared to WT genotypes, suggesting Piezo1 supports normal proliferation at this stage (*Figure 7B*). There was no significant difference in the percentage of cells in M-phase (PH-H3), and negligible evidence for cell death in either genotype (activated caspase three or pyknotic nuclei < 0.1%) (*Figure 1C*, *Table 1*), we therefore concluded the decreased proliferation was likely due to lengthening of cell cycle interphase.

We next tested several stages of early differentiation. When 'naïve' mESC cultures were converted to the next developmental stage, 'primed,' epiblast-like stem cells (EpiSC), no change in

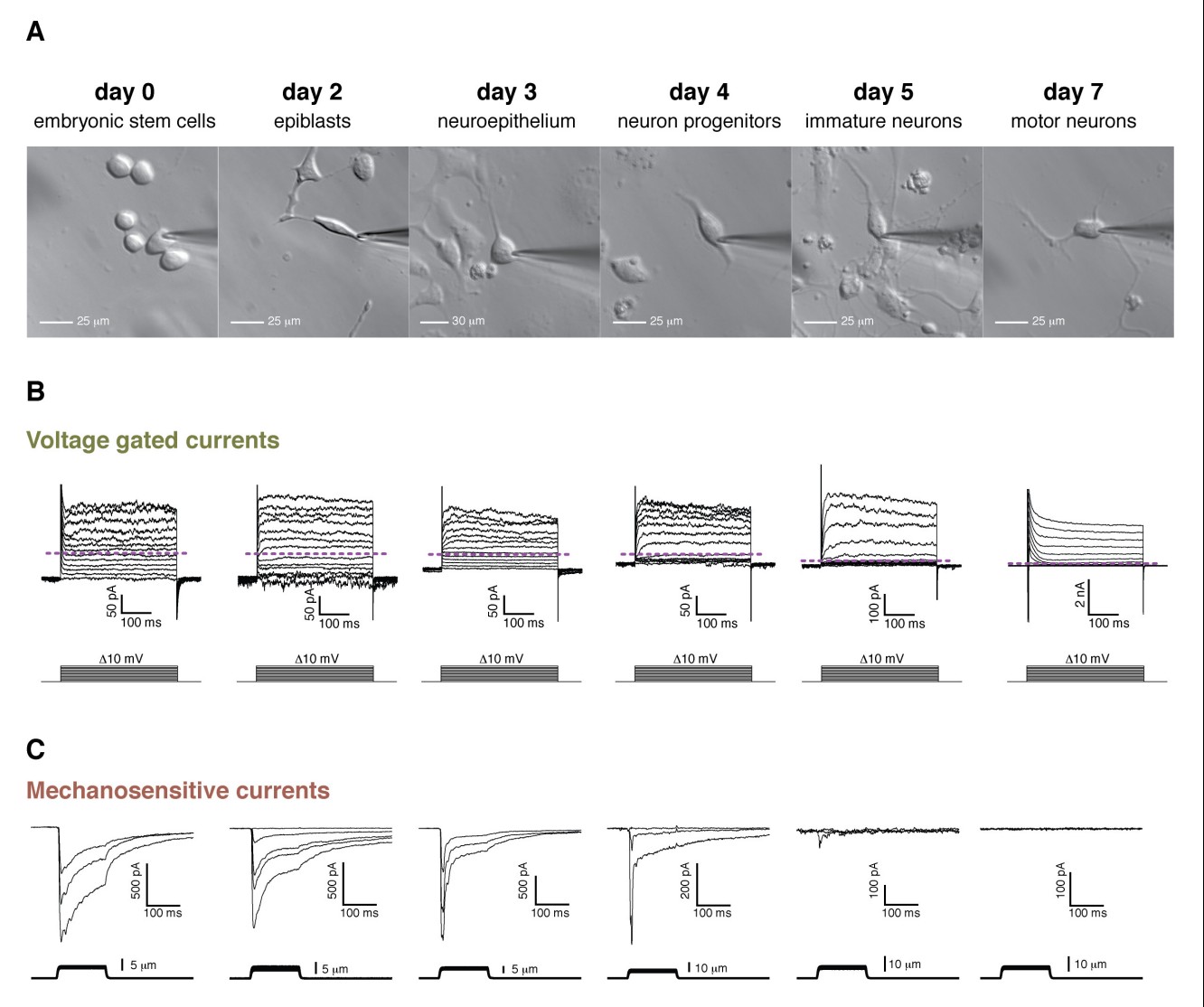

**Figure 3.** Cell morphology, voltage-gated currents, and mechanosensitive currents observed throughout the differentiation of mouse embryonic stem cells into motor neurons. (**A**) Images of the six developmental stages studied under brightfield illumination. (**B**) Voltage-gated currents were obtained in whole-cell mode by clamping the cells at −80 mV and depolarizing in steps from −80 mV to +40 mV. Examples for each stage are shown. The purple dashed line indicates the zero current level. Voltage-gated currents are relatively small in embryonic stem cells and in the first stages of the differentiation. Voltage-gated currents increase substantially in the later stages (notice the 10-fold difference in scale), pointing at the developing neuronal phenotype. Note, in day 7, the presence of very large rapid inactivated sodium currents at the beginning of the stimulation, immediately following the capacitive transient. (**C**) Mechanosensitive currents were obtained in whole-cell mode by clamping at −80 mV and poking at increasing depths. Examples for each stage are shown. These currents are heterogeneous but overall diminish in amplitude and become faster as the differentiation progresses.

DOI: https://doi.org/10.7554/eLife.33149.004

marker expression or colony organization was observed between genotypes (*Figure 7A,D*). Gastrulation was triggered by presentation of the in vivo cue BMP4, and differentiation to multiple germ lineages (SOX2, ectoderm; Brachyury, mesoderm) was not significantly different between genotypes (*Figure 7A,D,E*, *Table 1*). Finally, we asked if Piezo1 affected mESC response to substrate stiffness. Wild type mESC showed a lower rate of spontaneous differentiation under pluripotency conditions when cultured on a soft (28 kPa, PDMS) substrate compared to standard tissue culture plastic (100,000 kPa) following previous reports (*Chowdhury et al., 2010*). Piezo1 knockout clones showed the same relationship, indicating no effect of Piezo1 in mediating this response to substrate stiffness

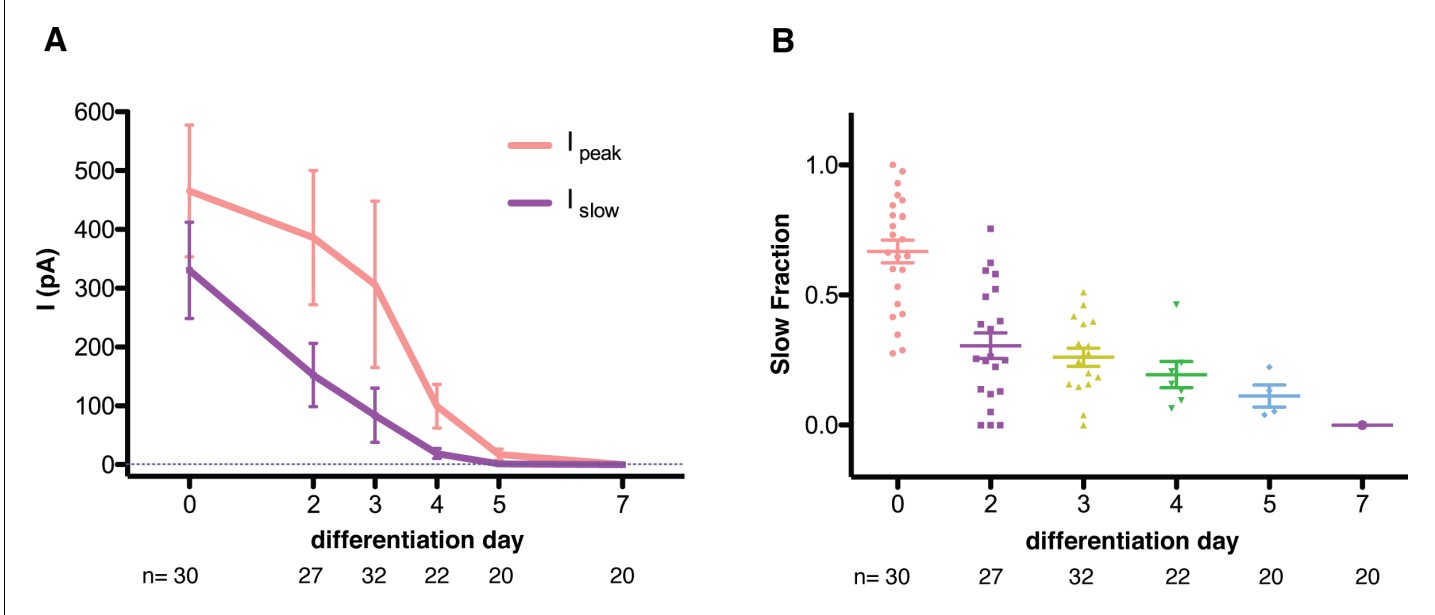

**Figure 4.** Quantification of mechanosensitive currents throughout the differentiation. (**A**) The evolution of the peak- and slow- currents throughout the differentiations (three independent differentiations were performed, and data were pooled). Peak-current is the maximum mechanosensitive current achieved by the stimulation. Slow-current is the mechanosensitive current measured 75 ms after the beginning of the stimulation. (**B**) The evolution of the slow fraction, defined as the ratio between slow and peak current, throughout the differentiation. The mechanosensitive current becomes smaller and faster as the differentiation progresses. Below each dataset is the total number of cells assessed at each stage.

DOI: https://doi.org/10.7554/eLife.33149.005

(*Figure 7F*). We conclude that while Piezo1 is expressed during pluripotency and mediates normal proliferation rate, it does not grossly affect the pluripotency per se, early germ layer differentiation, or response to substrate stiffness.

## Discussion

An initial observation of this study is that mES cells have large MS currents. These currents exhibit single channel conductance and cation selectivity similar to Piezo1 channels recorded in other cells under similar ionic conditions, and depend on the *Piezo1* gene. The identity of this MS current is therefore unequivocally assigned to Piezo1. Less clear is why embryonic stem cells would express high levels of a mechanosensitive ion channel. Until recently, growth factors were the most recognized factor influencing stem cells homeostasis and differentiation. However, recent reports have begun to show that mechanical cues are also able to sharply affect stem cell fate. For example, mechanical properties of the growth substrate can modulate the lineage choice of neural stem cells (*Pathak et al., 2014*). Other multipotent stem cells, like mesenchymal stem cells, have long been known to be influenced by matrix elasticity (*Engler et al., 2006*). Although the action pathway of these variables is not fully understood, the presence of mechanosensitive ion channels such as Piezo1 suggests a role for MS channels as potential transduction molecules in stem cells.

Another derivation of this study is that Piezo1 forms a slowly-inactivating MS current in mES cells. Having excluded the possibility of an intrinsically slow inactivating channel by virtue of its amino acid sequence, we are left to assume that Piezo1 is somehow regulated to have slow inactivating kinetics. Regulation of its kinetics can arise from either additional components (e.g. beta subunits) or from mechanical properties of the particular cellular environment, or a combination of both. It is interesting to note that the mRNA expression of *Piezo1* throughout the differentiation of mES cells into motor neurons (*Figure 5*) correlates with the fast component of the MS current. The total MS current observed follows a slightly different evolution (*Figure 4*). In light of the results of this work, it is foreseeable that there could exist additional components that shape the Piezo1 currents throughout the differentiation to modify their amplitude and kinetics into their final observed form. In summary,

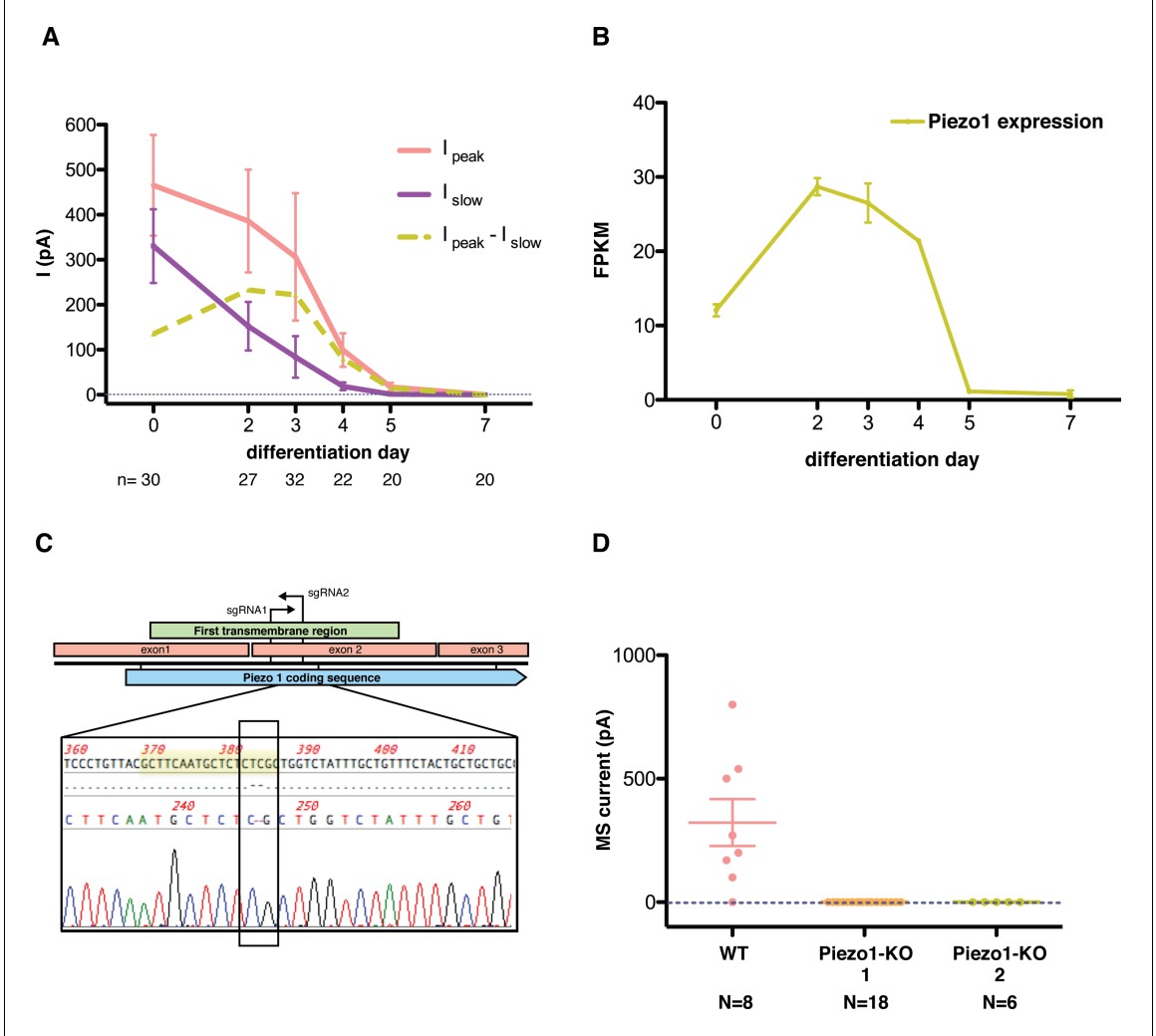

**Figure 5.** Piezo1 throughout the differentiation of embryonic stem cells into motor neurons. (A) Peak- and slow- mechanosensitive currents throughout the differentiation are plotted together with the difference between them, which we called 'fast component'. (B) *Piezo1* expression throughout the differentiation as assessed by transcriptome analysis. The expression of *Piezo1* throughout the differentiation strikingly resembles the evolution of the fast component of the mechanosensitive current. FPKM: fraction per kilobase per million reads. (C) Schematic of the Crispr design. The diagram shows the beginning of the *Piezo1* mRNA in mES cells. The first three exons are shown, along with the coding sequence (CDS) and the first predicted transmembrane (TM) region. Two guide RNA sequences (sgRNAs) were chosen to generate a double strand break in the beginning of the first TM region. Below the diagram, a sequence reaction of a fragment of DNA extracted from one of the modified colonies is shown. In yellow is marked the targeted sgRNA sequence, and boxed in black is marked the region with a two base-pair deletion that generates a frame-shift mutation, and an early stop codon shortly after. Only one sequence for each colony was obtained after sequencing with no background, indicating a homozygous mutation. (D) Both Piezo1 knock-out colonies of mouse embryonic stem cells showed no mechanosensitive activity.

DOI: https://doi.org/10.7554/eLife.33149.006

these results point to a modification of the Piezo1 protein or the environment in which it functions, which alters the Piezo1 gating properties. These results advance our understanding in two respects. First, they point to a role for Piezo1 in stem cell function, and second, they suggest that an additional regulatory mechanisms that shape the gating properties of Piezo1.

We still know very little about modulation of Piezo1 behavior. In a recent work, Bae et al showed that the kinetics of human PIEZO1 can be regulated by pH (*Bae et al., 2015*). Additionally, Sack's group observed that in certain conditions a fast inactivating Piezo1 channel can be 'converted' into a slow one by repeated stimulation (*Gottlieb et al., 2012*). They postulate that Piezos could be located in confined arrays ('corrals') that can be disrupted through strenuous stimulations, and that the gating mechanism is somehow linked to these spatial arrays. Protein modulators of Piezo1 have

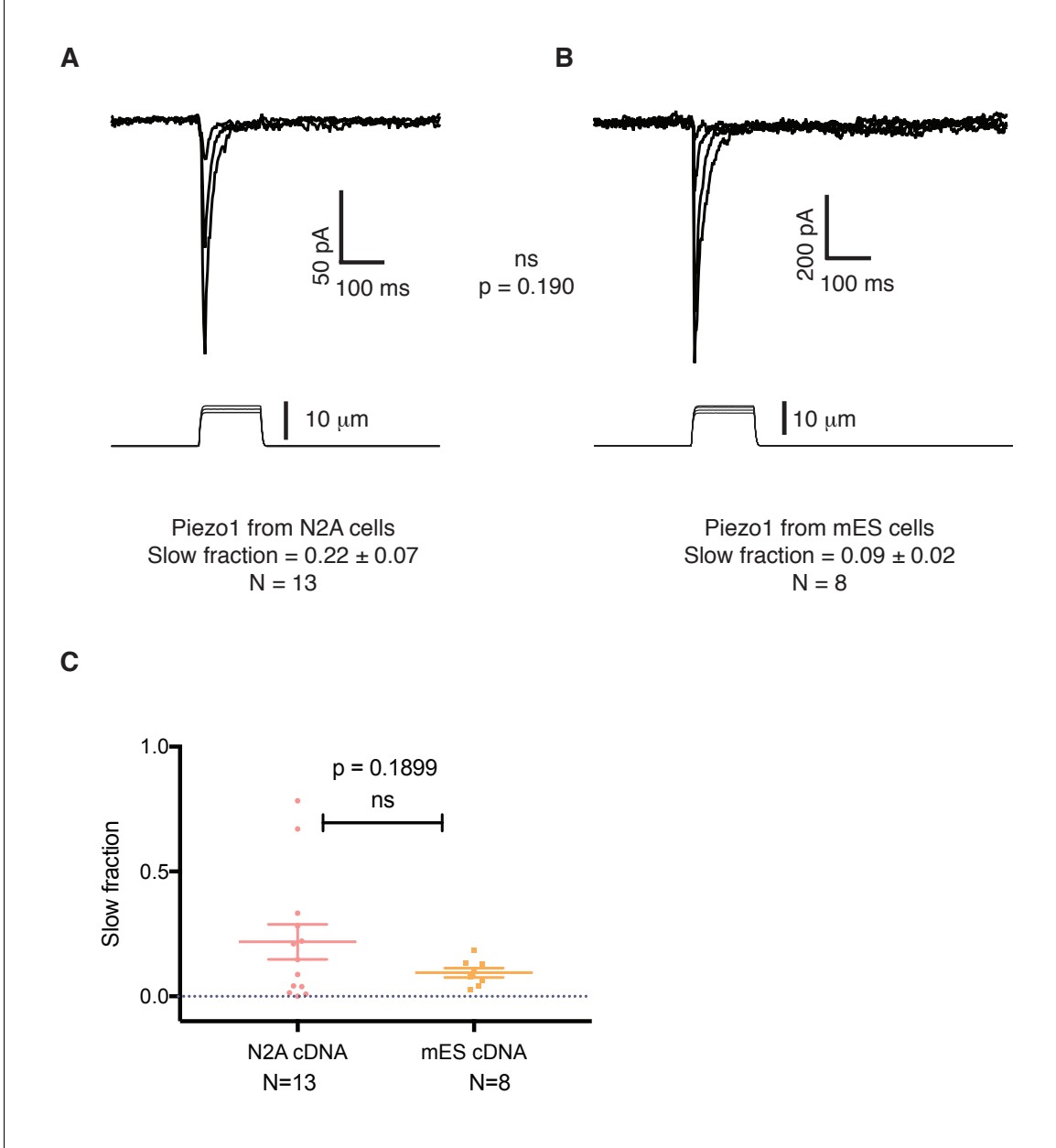

**Figure 6.** The slower kinetics of Piezo1 in mouse embryonic stem cells are not a result of mutations in its coding sequence. (**A**) Mechanosensitive currents elicited by expression of Piezo1 cDNA from N2A cells in HEK293 cells. (**B**) Mechanosensitive currents elicited by expression of Piezo1 cDNA from mouse embryonic stem cells in HEK293. Both cDNAs differ in three aminoacids. (**C**) Slow fraction of mechanosensitive currents in HEK293 cells expressing either Piezo1 cDNA from N2A cells (left) or from mES cells (right). Expression of both cDNAs in HEK293 cells shows no significant difference in the kinetics, p=0.1899.

DOI: https://doi.org/10.7554/eLife.33149.007

also been found: Stoml3, the mammalian counterpart of MEC-2, is required for mechanosensation in ~40% of myelinated mechanosensory fibers (*Poole et al., 2014*). This protein is reported to tune up the threshold of sensitivity of Piezo2 currents in DRGs, such that its absence increases the threshold of activation of Piezo2 by an order of magnitude. The effect on Piezo2 is not specific: Piezo1 currents can too be increased and their thresholds lowered by expression of Stoml3. These results were reproduced by heterologous expression of Stoml3 with either Piezo1 or Piezo2 in HEK293 cells. Additionally, Stoml3 protein co-precipitates with Piezo1 or Piezo2 after heterologous expression in

HEK293 cells, indicating some degree of association. Polycystin-2 (PC2) is another proposed modulator of Piezo1 currents (*Peyronnet et al., 2013*). Overexpression of this protein reduced endogenous Piezo1 currents in proximal convoluted tubule cells and in overexpression experiments in COS cells. PC2 also co-precipitates with Piezo1 when overexpressed in COS cells.

Lastly, it is likely that other unknown mechanosensory ion channels exist. The Patapoutian group reported that only the fast-inactivating component of the mechanosensory currents in DRG neurons can be attributable to Piezo2 (*Coste et al., 2010*). Piezo1 is not supposedly expressed in those cells, so we are left to assume that there is still at least one novel source of slow- and/or intermediate-inactivating mechanosensitive currents present in DRGs. A most fascinating challenge in the field will be to identify the source of that current and unveil the mechanisms behind its behavior and function.

Overall Piezo1 KO ES cell lines showed minimal changes in pluripotency or differentiation in vitro compared to wild type lines. This finding fits with the normal early development reported for Piezo1 KO mice. However, outside of the lethal ~e14 vascularization defect reported for these mice, little specific or functional phenotyping was done, leaving open the possibility of subtle functional defects occurring early in development, which may be compensated by homeostatic developmental mechanisms. Based on functional channel expression during pluripotency and early differentiation, we originally hypothesized that Piezo1-mediated mechanoreception in epiblast cells would affect early development. This hypothesis was based on the crucial but poorly defined role for mechanical forces in the tissue morphogenesis, which establishes and defines gastrulation, as well as in vitro studies using mouse and human ES cells (*Chowdhury et al., 2010*; *Sun et al., 2014*; *Przybyla et al., 2016*). Our findings failed to directly support this hypothesis, but they do not exclude a role for Piezo1 mediated mechanotransduction in peri-gastrulation development. This is partly due to the limitations of established and informative in vitro stem cell models of gastrulation, especially the lack of stereotypical in vivo egg cylinder architecture, which suggests a role for mechanical feedback in the first place. It is therefore possible that as more sophisticated, reproducible and in vivo-like in vitro models are developed (*Morgani et al., 2018*, e.g.) a role for Piezo1 at this stage may yet be uncovered in vitro, but further in vivo or ex vivo approaches will likely be more relevant at present. The proliferation activity we identified could well play a role in normal gastrulation, by regulating the size of the epiblast, therefore influencing the functional geometry of the egg cylinder and ectoplacental cone, which triggers and organizes gastrulation. If so, the viability of Piezo1 KO mice argues that this activity must be almost fully compensated for by an unknown homeostatic mechanism, which could be identified in the future.

## Materials and methods

### Cells

HEK293 tsA201 cells were obtained from Sigma and maintained in DMEM (Gibco) supplemented with 10% Fetal Bovine Serum (Gibco) and 1% L- Glutamine (Gibco). Mouse embryonic stem cells (Hb9-GFP) were obtained from the Wichterle lab (*Wichterle et al., 2002*). This cell line contains a GFP transgene driven by the Hb9 promoter, a motor neuron specific promoter. Cells were kept in serum-free 2i + LIF media (*Ying et al., 2008*) and passaged every 2–3 days.

For electrophysiological recordings of mouse embryonic stem cells, cells were plated on 12 mm poly-D-lysine coated coverslips (NeuVitro), pre-coated with Matrigel (Invitrogen). Mouse embryonic stem cells normally grow forming tight associations called embryonic bodies that prevent patch clamp procedures. In order to facilitate electrophysiological recordings, cells were dissociated to single-cell level using Accutase (Gibco) or Trypsin (Gibco) and plated at low density (2,000 to 25,000 cells per 12 mm coverslip) in the presence of Rho-K inhibitor (Millipore). MS currents were recorded ~4–10 hr after plating. For electrophysiological recordings of Day 5 and Day 7 cells, coverslips were coated with PDL and Laminin EHS.

For differentiating mouse embryonic stem cells into motor neurons we followed protocols by the Wichterle lab (*Wichterle et al., 2002*), but replacing Sonic Hedgehog by SAG (smoothened agonist). Appearance around Day 5 of GFP-positive neurons signals a successful specification of motor neurons. The differentiation was performed three independent times and the electrophysiological results of all three differentiations were pooled together.

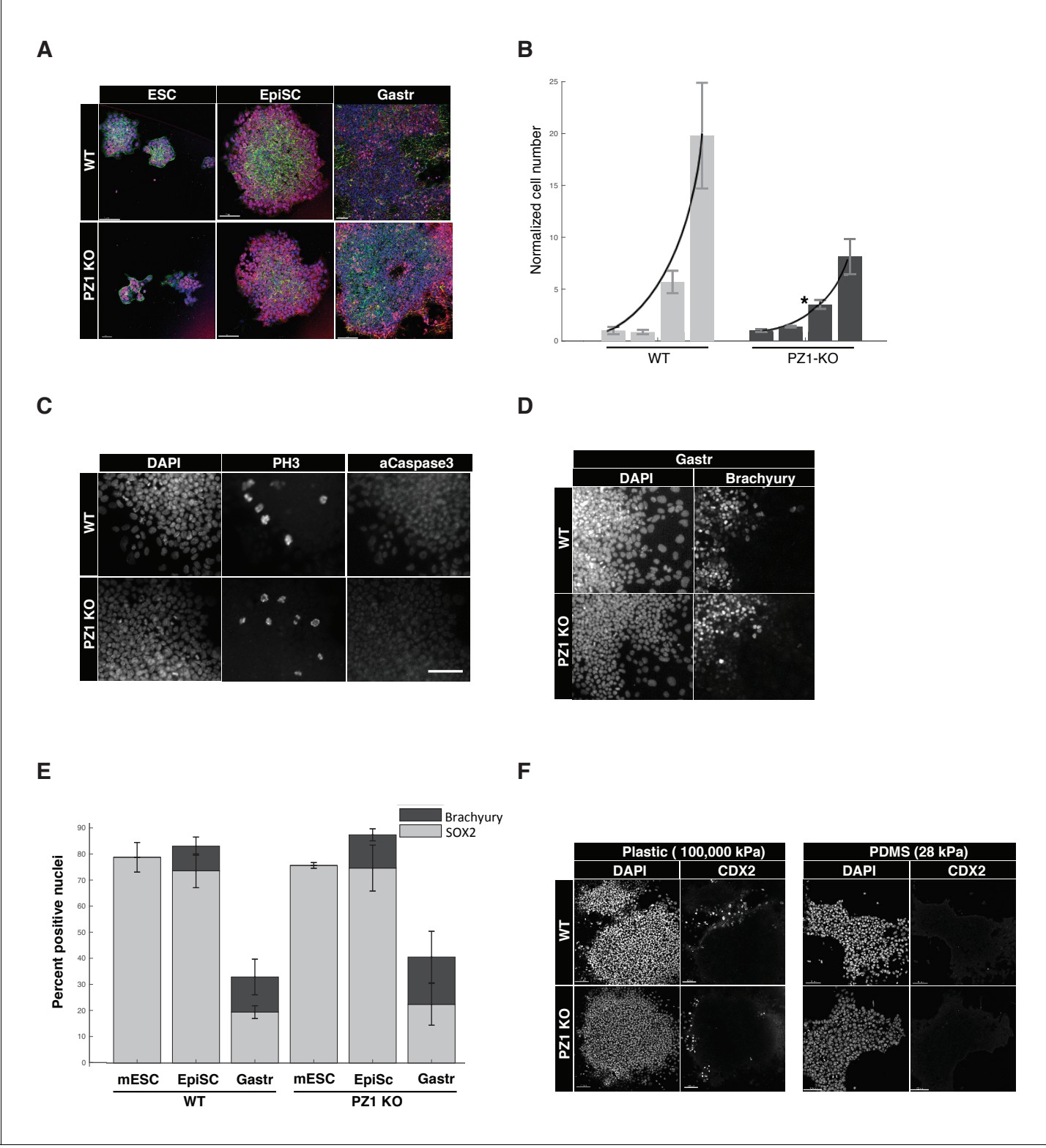

**Figure 7.** Piezo1 knockout affects the rate of proliferation but not pluripotency or gastrulation phenotypes in mouse ESCs. (**A**) Immunostaining and confocal imaging of Piezo1 knockout (PZ1KO) pluripotent stem cells (bottom row) at naïve (ESC), primed (EpiSC), or gastrulation-model (Gastr) stages shows no differences in expression of pluripotency master regulator SOX2 (pink), typical pattern of cell to substrate (focal adhesions, at lower z, not shown) and intercellular adhesion by F-Actin (Phalloidin, green) and E-Cadherin (red), or general colony morphology. 1.6 μm Z section or oblique virtual section of representative colonies of WT and Piezo1 knockout (n = 2,3) ESC clones. 20x z stacks, scale bar 80 μm. (**B**) Growth curve of WT mESC stage
*Figure 7 continued on next page*

Figure 7 continued

cells showed PZ1KO lines were significantly slower (exponential curve fit). Mean number of cells per well per genotype ± sd (n = 2 WT, 3 KO lines per genotype) normalized to mean per genotype at day1. Exponential fit constant significantly different Students T-test p<0.05. (C) No significant difference in mitotic fraction or cell death. Immunostaining and epifluorescent imaging of WT (top) and PZ1KO for DAPI, M-phase cells (PH3), and dying cells (Caspase 3, pyknotic nuclei) showed no significant difference between genotypes in mitotic fraction (PH3 +nuclei) mean/genotype ± sd%, WT, n = 2, PZ1KO, n = 3 lines. Neither the cell death marker activated caspase3 or pyknotic nuclei were detectable in either genotype, confirming equivalently negligible levels of cell death. Scale bar is 80μm. (D) Epifluorescent imaging of transcription factor markers of pluripotency (SOX2, *Figure 1A*) and ectoderm (SOX2, *Figure 1A*) and Mesoderm (Brachyury) germ layer identities at naïve (mESC), primed (EpiSC) and gastrulation stage differentiation (Gastr, shown). Same scale bar as in C. (E) Quantitative cell scoring showed no significant difference between WT (n = 2) and PZ1KO (n = 3) clones, mean% of cells/line+/sd at each stage. Brachyury not tested at mESC stage since no differentiation was present. Decreased SOX2 percentage at Gastr stage indicated restriction of SOX2 expression to ectodermal lineage during gastrulation and was not different between genotypes. T-test p<0.05. (F) EpiSC stage cultures grown on standard tissue culture plastic substrate (100,000 k Pa) showed a small subpopulation of spontaneously differentiating cells (CDX2+, trophectoderm, embryonic endoderm, or paraxial mesoderm) whereas culture on a less stiff substrate (28 kPa PDMS) showed no spontaneous CDX2 differentiation. PZ1KO lines showed the same differential differentiation inhibiting response to substrate stiffness.

DOI: https://doi.org/10.7554/eLife.33149.008

## Electrophysiology

All poking experiments were performed using a probe drawn from borosilicate glass (Sutter Instruments) fire polished (MF-83, Narishige Co.) until sealed. The probe was mounted to piezo-driven actuator driven by a controller/amplifier (P- 601/E-625; Physik Instrumente) controlled through Clampex software. After formation of a whole-cell seal by a different electrode, the probe was positioned at 60° to the cell ~2 μm away from the membrane.

All pressure applications through patch pipettes were performed with a high- speed pressure clamp (ALA Scientific) controlled through Clampex software. Pressure application velocity was set to the maximum rate of 8.3 mmHg/msec.

Whole-cell recordings were performed using extracellular solution (mM): $150NaCl$, $2MgCl_2$, $3KCl$, $2CaCl_2$, $10Hepes$, $10Glucose$ (pH 7.4 with NaOH, 325 mOsm/kg) and intracellular solution (mM): $150KCl$, $10EGTA$, $10Hepes$, $1EDTA$ (pH 7.4 with NaOH, 310 mOsm/kg). Electrodes were drawn from borosilicate patch glass (Sutter Instruments) and polished (MF-83, Narishige Co.) to a resistance of 3–6 MOhms. Analog signals were filtered (1 kHz) using the built-in 4-pole Bessel filter of an Axopatch 200B patch clamp amplifier (Molecular Devices) in patch-mode and digitized at 20 kHz (Digidata 1440A, Molecular Devices). The access resistance was monitored throughout the recordings to avoid incurring in excess series resistance, and care was put into keeping the series resistance low by either recording from cells with overall low levels of current (under 1nA) and using pipettes of small initial resistance.

For single channel studies the following solutions were used (mM): Pipette: $150KCl$, $10Hepes$, $10EGTA$ (pH 7.3 with NaOH, 310 mOsm/kg). Bath: $150NaCl$, $3KCl$, $2CaCl_2$, $2MgCl_2$, $10Hepes$, $10Glucose$ (pH 7.3 with NaOH, 310 mOsm/kg). All recordings were done in excised outside-out mode. Analog signals were filtered (1 kHz) using the built-in 4-pole Bessel filter of an Axopatch 200B patch clamp amplifier (Molecular Devices) in patch- mode and digitized at 20 kHz (Digidata 1440A, Molecular Devices).

For ion selectivity studies the following solutions were used (mM): Intracellular; $150KCl$, $10Hepes$, $10EGTA$ (pH 7.3 with KOH, 310 mOsm/kg). Extracellular NaCl; $150NaCl$, $10Hepes$, $10Glucose$ (pH 7.3 with NaOH, 310 mOsm/kg). Extracellular NaGluconate; $152NaGluconate$, $10Hepes$, $10Glucose$ (pH 7.3 with NaOH, 310 mOsm/kg). Extracellular NMDG-Cl; $152NMDG-Cl$, $10Hepes$, $10Glucose$ (pH 7.3 with KOH, 310 mOsm/kg). Extracellular $CaCl_2$; $90CaCl_2$, $7.5Hepes$, $7.5Glucose$ (pH 7.4 with NaOH, 310 mOsm/kg).

## Transcriptome analysis

Total RNA was extracted at days 0 (mouse embryonic stem cells), 2, 3, 4, 5, and 7 (motor neurons) of each differentiation using a Trizol/RNeasy hybrid protocol. Briefly, cells are homogenized using a recommended volume of Trizol (Invitrogen), chloroform is added in the recommended volume and the mix is shaken vigorously. After centrifugation to allow phase separation, the aqueous phase is kept and mixed with 70% ethanol 1:1. The protocol follows using an RNeasy column (QIAGEN) and following the manufacture's instructions. For days 5 and 7 a step was added previous RNA

**Table 1.** Quantitative comparison of pluripotency and differentiation phenotypes.

Proliferation was measured by seeding 4 96well plate wells, imaging whole wells at day 0, 1, 3, and 6, and scoring images for total cells (DAPI+) and subtracting the number of Sytox orange+ (dead) cells = total live cells. Total growth: table shows mean cells/line/genotype/day and standard deviation; n = 2 and n = 3 lines per WT or KO genotype respectively. Total cells is the sum of cells in four replicate wells/line. Normalized growth: Each line was normalized to its own day 0 value. Curve Fit Parameters: each line was fit with an exponential growth curve and the mean growth constants were computed and found not significantly different. Colony Size: whole colonies were segmented using MultiWavelength Cell Scoring adjusting the size and intensity thresholds so that entire colonies were segmented by DAPI signal, rather than individual cells. Mean values for n colonies counted for each line at day 0 and day six are shown, unit is $\mu m^2$. No significant differences were found between genotype at either day. Proliferation: Three randomly chosen 10x fields (mean 1473 cells/field) per line were scored for total DAPI +cells and cells positive for phosphohistone H3 (PH3) in Metamorph. No significant difference between genotypes was observed. No cells were positive for activated caspase-3 or showed pyknotic nuclei, indicating negligible levels of cell death across genotypes. All statistics Student's t-test, 1 tail, 2 sample unequal variance.

| | Total growth | | | |
|---|---|---|---|---|
| | WT (n = 2) | SD | KO (n = 3) | SD |
| Day 0 | 402.50 | 127.01 | 721.42 | 115.63 |
| Day 1 | 342.75 | 61.23 | 849.33 | 64.51 |
| Day 3 | 2289.50 | 334.41 | 2043.17 | 346.46 |
| Day 6 | 7919.38 | 1728.79 | 5234.17 | 1350.08 |
| | **Normalized growth** | | | |
| | WT (n = 2) | SD | KO (n = 3) | SD |
| Day 0 | 1.000 | 0.356 | 1.000 | 0.133 |
| Day 1 | 0.864 | 0.206 | 1.388 | 0.077 |
| Day 3 | 5.689 | 1.076 | 3.518 | 0.427 |
| Day 6 | 19.802 | 5.096 | 8.124 | 1.708 |
| | **Curve Fit Parameters** | | | |
| | Growth Constant | Goodness of Fit | | |
| HB9 (wt) | 0.443 | 0.985 | | |
| WT13 | 0.489 | 0.996 | | |
| KO5 | 0.377 | 0.993 | | |
| KO6 | 0.287 | 0.994 | | |
| KO8 | 0.316 | 0.971 | | |
| | Mean Growth Constant | SD | | |
| WT | 0.326 | 0.033 | | |
| KO | 0.466 | 0.046 | | |
| | p-value | 0.316 | | |
| **Colony size** | | | | |
| | Day 0 | | Day 6 | |
| | Mean | SD | Mean | SD |
| HB9 (wt) (day 0, 6: n = 122, 30) | 45666.2 | 81641.1 | 133231.9 | 66024.3 |
| WT13 (day 0, 6: n = 74, 28) | 78107.7 | 94966.6 | 156237.9 | 90772.1 |
| KO5 (day 0, 6: n = 128, 24) | 40269.1 | 74822.2 | 129713.2 | 81019.6 |
| KO6 (day 0, 6: n = 69, 17) | 101383.7 | 120527.3 | 95029.8 | 42050.6 |
| KO8 (day 0, 6: n = 71, 19) | 84821.1 | 108565.5 | 145450.6 | 107648.8 |
| | Day 0 | | Day 6 | |

*Table 1 continued on next page*

|  | Mean | SD | Mean | SD |
|---|---|---|---|---|
| WT | 61886.9 | 22939.7 | 144734.9 | 16267.7 |
| KO | 75491.3 | 31607.4 | 123397.8 | 25796.9 |
| p value | 0.618 |  | 0.170 |  |
| **Proliferation** |  |  |  |  |
|  | % PH3+ of total DAPI |  | mean% PH3+ | SD |
| HB9 (wt) | 3.487 | WT | 3.224 | 0.372 |
| WT13 | 2.961 | PZ1KO | 3.424 | 0.218 |
| KO5 | 3.331 |  | T-test p value | 0.292 |
| KO6 | 3.268 |  |  |  |
| KO8 | 3.673 |  |  |  |

DOI: https://doi.org/10.7554/eLife.33149.009

extraction: cells were sorted using a BDFACSAria Cell Sorter (BD) to isolate only the GFP-positive fraction, therefore enriching the sample in motor neurons. GFP-positive cells were sorted into Trizol LS (Invitrogen) and extraction of RNA proceeded as before.

High throughput RNA sequencing was done in triplicates except for days 5 and 7, which were done in duplicate. Each duplicate or triplicate sample was obtained from an independent differentiation. The RNA samples were first treated with DNase, then one library per sample was prepared using Illumina's TruSeq RNA Sample Prep Kit, where polyA-fragments were selected, followed by cDNA synthesis and ligation of amplification and sequencing adapters. Libraries were then individually barcoded and then pooled with six libraries per lane on an Illumina HiSeq 2500 instrument (Illumina). All samples were sequenced as single-read with read lengths of 50 bp. For analysis of RNA-seq data, reads were uploaded to the Galaxy environment (usegalaxy.org) and were curated and trimmed according to the quality of the sequences using default options. Curated sequences were ran through Tophat for Illumina using a built- in reference genome mm10 (GRCm38/mm10) for mapping the reads to the mouse genome. Sequences were then ran through the Cufflinks package which assembles transcripts and estimates their abundances to obtain, for each sample, a list of transcripts with their associated transcript counts. Transcript counts are obtained as FPKMs (fragments per kilobase of transcript per million mapped reads). Finally, Cufflinks (CuffDiff) was used to calculate differential expression of transcripts between samples.

High-throughput data is publicly available at the GEO database, for public access (accession number GSE106526).

## Crispr-CAS9 knockout of Piezo1

px459 (Addgene) was used to express Cas9 and guide RNA sequence along with a Puromycin resistance cassette according to published procedures (*Ran et al., 2013*). Two guide RNA sequences were cloned separately to obtain two independent knock-out colonies. Sequences were ACGC TTCAATGCTCTCTCGC and AGAGAGCATTGAAGCGTAAC, both located in the beginning of the second exon of the mouse *Piezo1* gene. hB9-GFP mES cells were then transfected with the px459 vector using Lipofectamine 2000 (Invitrogen) and selected with 1 µg/ml Puromycin for 2 day. After 1 day of recovery in Puromycin-free medium, transfected cells were dissociated and sparsely re-plated into 10 cm dishes. Single colonies were isolated after 7 days, expanded, and DNA was extracted using QuickExtract DNA Extraction solution (Epicentre). A 500 bp region containing the Cas9 target was amplified by PCR using the following primers: CGTGTGCATCCACGTATGA and AGGTG TGCACTGAAGGAACC. Obtained fragment was then sequenced. Sequencing results showed that some colonies contained a mix of 2 sequences, indicating differential mutations in both alleles of the *Piezo1* gene. However, four colonies showed a clear single sequence, indicating a homozygous mutation near the PAM sequence. Two of those colonies were selected for electrophysiological studies. A third colony was added to the studies for figure 7.

## Cloning of Piezo1 cDNA from mouse embryonic stem cells

Total RNA was extracted from mES cells using RNeasy kit (QIAGEN) and cDNA was made using Quantitect Reverse Kit (QIAGEN). The following primers were used in various combinations to obtain PCR fragments that cover the entire coding region of the mouse *Piezo1* gene: TGCACTACTTCCA-CAGACCG, CAGGAAGATGAGCTTGGCGT, CTACTCCCTCTCACGTGTCCA, TCTACTGGCTGTTGC TGCC, CCAGCAACACAATGACCAGC, ATGGAGCCGCACGTGCTG, GATGCTGCCCCAGCCG TGGG, GGCCTGCCTCATCTGGACGG, AGCAGTTGGGCGACCTGGGC, TGCCCGCCCAGGCTGTG TGC, AGCCCAGCTCGTGCTGTGGG, CACGGTAGACGGGCTGACGC, CGGCGCTATGAGAA-CAAGCC, CGACCGTGCCCTCTACCTGC, GGAGTATACTAATGAGAAGC, AGG GACGCTGTG TCCCTACC, TACTGGATCTATGTGTGCGC, CATACCAGGTCACACAGGTC, TCCTCCTGATGC TCAAGCAGAGG, CTAGGTCCAGCAGCCGGTCAG, CTCACTCCATCATGTTCGAGG. PCRs were done using Phusion HF (NEB) or Pfu Ultra II (Agilent). For some difficult reactions 5% DMSO was added to the PCR reaction. The Piezo1 construct from N2A cells was obtained from the Patapoutian lab. For whole cell poking of both constructs transfected into HEK293 cells the following solutions were used (mM): 150KCl, 10Hepes, 10EGTA (pH 7.3 with NaOH, 310 mOsm/kg; intracellular) and 150NaCl, 3KCl, 2CaCl$_2$, 2MgCl$_2$, 10Hepes, 10Glucose (pH 7.3 with NaOH, 310 mOsm/kg; extracellular).

## ESC phenotyping and data analysis

ESC proliferation was measured by seeding in 250 ESC per well in 96 well plates coated with 5 ug/ ml laminin5,2,1 (biolamina). Wells were stained with markers for all nuclei (Hoechst, NucBlue, Thermo) and dead cells (SytoxOrange, Thermo) and whole wells were imaged and quantified for live cells on days 0, 1, 3 and 6 using ImageExpress and Metamorph. 'Naïve' (Inner cell mass-like) ESC were maintained in '2i + LIF medium' and transitioned to 'primed' (epiblast-like stem cells) EpiSC by incubation for 48 hr with mouse embryonic fibroblast conditioned medium with 20% ES cell grade FCS (CM) containing activin (from feeders) and added bFGF (20 ng/ml, Thermo) without LIF or 2i used for naïve culture, and germ layer differentiation was triggered by adding 50 ng/ml human recombinant BMP4 (R & D) to CM and incubating for 48 hr. Cultures for staining were seeded as above on Ibidi uclear eight well slides, or 28 kPa PDMs dishes (Ibidi), fixed with 4%PFA, permeabilized with 0.1%Triton, stained with the following primary antibodies (goat SOX2 1:500, #AF2018-SP, R and D Systems; rabbit E-Cadherin 1:200, #3195S, Cell Signaling Technologies; mouse N-Cadherin 1:100, #350802, BioLegend; rabbit SOX2 1:200, #3579S, Cell Signaling Technologies; goat Brachyury, 1:300, AF2085, R and D Systems; mouse CDX2 1:200, CDX2-88, BioGenex; phalloidin-647 1:50, A22287, Life; rabbit phospho-histone (S28) H3 1:500, #4178, Abcam; rabbit cleaved Caspase3 1:400, #9664S, Cell Signaling Technologies) and secondary antibodies (Donkey anti-mouse/rabbit/ goat Alexa 488 555 647, Invitrogen) and DAPI (Life), imaged on a Leica SP8 inverted confocal with gallium arsenide detectors at 12 bits, 1024 × 1024 pixels using an HCX PL APO CS 20x/0.75NA air objective at 1 Airy unit pinhole and Nyquist spacings. Raw image data was analyzed quantitatively by unbiased automated image analysis, Metamorph MultiWavelength Cell Scoring module, and qualitatively by 3D blind deconvolution (Autoquant) 10 iterations and 3D-rendering (Imaris, Bitplane), contrast and gain were adjusted to maximize visual legibility, but kept constant across all fields and lines from the same stain.

## Acknowledgements

We thank Hanbin Wang and Cecilia Pellegrini for assistance with stem cell culture and differentiation, and the Rockefeller University Bio Imaging Resource Center for imaging and analysis. We thank members of the MacKinnon laboratory for helpful discussions and especially Ernest Campbell, Xiao Tao and Steve Brohawn for their insight. RM is an investigator in the Howard Hughes Medical Institute. JIdM was supported by a Howard Hughes Medical Institute International Student Fellowship.

## Additional information

### Funding

| Funder | Author |
|---|---|
| Howard Hughes Medical Institute | Josefina Inés del Mármol<br>Roderick MacKinnon |

The funders had no role in study design, data collection and interpretation, or the decision to submit the work for publication.

### Author contributions

Josefina Inés del Mármol, Conceptualization, Data curation, Formal analysis, Validation, Investigation, Visualization, Methodology, Writing—original draft, Writing—review and editing; Kouki K Touhara, Validation, Investigation, Writing—review and editing; Gist Croft, Data curation, Formal analysis, Investigation, Methodology, Writing—review and editing; Roderick MacKinnon, Conceptualization, Resources, Formal analysis, Supervision, Funding acquisition, Methodology, Project administration, Writing—review and editing

### Author ORCIDs

Josefina Inés del Mármol http://orcid.org/0000-0003-2715-0482
Roderick MacKinnon http://orcid.org/0000-0001-7605-4679

### Decision letter and Author response

Decision letter https://doi.org/10.7554/eLife.33149.014
Author response https://doi.org/10.7554/eLife.33149.015

## Additional files

### Supplementary files

• Transparent reporting form
DOI: https://doi.org/10.7554/eLife.33149.010

### Data availability

Sequencing data have been deposited in GEO under accession number GSE106526. Source dat files have been provided for figures 1, 2, 4, 5, and 6, and source data for figure 7 is included as supporting file.

The following dataset was generated:

| Author(s) | Year | Dataset title | Dataset URL | Database, license, and accessibility information |
|---|---|---|---|---|
| del Mármol JI, Touhara KK, MacKinnon R | 2017 | High-throughput RNA seq data of mouse embryonic stem cells and intermediate states throughout their differentiation into motor neurons | https://www.ncbi.nlm.nih.gov/geo/query/acc.cgi?acc=GSE106526 | Publicly available at the NCBI Gene Expression Omnibus (accession no: GSE106526) |

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
