## [Decision Letter]

Thank you for submitting your article "Piezo1 forms a slowly-inactivating mechanosensory channel in mouse embryonic stem cells" for consideration by *eLife*. Your article has been reviewed by three peer reviewers, including Kenton J Swartz as the Reviewing Editor and Reviewer #1, and the evaluation has been overseen by Richard Aldrich as the Senior Editor.

Summary:

This is an interesting and well executed study showing that mouse embryonic stems cells express mechanosensitive ion channels that exhibit slowly desensitizing properties in response to poking with mechanically actuated probe. Measurements of whole-cell mechanosensitive currents with different cations in the extracellular solution suggest that the channels are relatively non-selective between cations, with a small preference for calcium over sodium or potassium. Single channel measurements with excised patches show similar kinetic properties when elicited with pressure-clamp and compared to whole-cell measurements obtained with the poking assay, and they show that the mechanosensitive channels have a single channel conductance of ~25 pS. When ES cells are differentiated into motor neurons the mechanosensitive currents become more rapidly desensitizing before they disappear, as voltage-gated conductances become larger. Interestingly, the authors show that expression of Piezo1 correlates with the amplitude of the mechanosensitive currents during differentiation, and KO of Piezo1 leads to loss of mechanosensitive currents. Finally, the authors clone Piezo1 from the ES cells and although they identify three residues differing from the previously cloned Piezo1 subunit, expression in mammalian cells gives rise to rapidly desensitizing currents that resemble those previously reported for the Piezo1 channel. The interesting take home message is that a regulatory mechanism must exist in ES cells that converts Piezo1 from rapidly desensitizing to slowly desensitizing. The manuscript was easy to read and comprehend and it makes a valuable contribution to the literature.

Although all three reviewers were enthusiastic about this study, the consensus was that the work would be more appropriate for *eLife* if the authors could provide evidence for what function Piezo1 might have in mES cells or for the mechanism by which extrinsic factors might be shaping Piezo1 kinetics in different cellular environments.

Essential revisions:

1) Piezo1 has known to be widely expressed, so the observation they are in mES is interesting, but perhaps not particularly surprising. More to the point, both Piezo1 and Piezo2 channels have already been shown to have different kinetics depending on the cell types where they are found. Piezo2-dependent currents in Merkel cells have a shoulder current not commonly observed in sensory neurons (Woo et al., 2014; Ikeda et al., 2014). Very recently, it was shown that Piezo2 mediates at least some of the slower adapting currents in duck sensory neurons (Schneider et al., 2017). Perhaps more to the point, the currents reported here look remarkably similar to those previously reported to be found in some types of HEK293 cells (Dubin et al., 2017).

2) While the authors discuss possible implications of Piezo1 function in mES cells (Discussion section), they do not attempt to test those hypotheses experimentally. This is puzzling given the authors took the time to create Piezo1 KO mES cells. There seems to be a missed opportunity to experimentally examine the role of mechanosensitive signaling in mES cells differentiation, outgrowth, survival etc. At a minimum, the authors could have tested their hypothesis about a role for Piezo1 in sensing substrate stiffness.

3) The authors do a nice job of showing that the mES mechanosensitive currents are mediated by Piezo1 (Figure 5D) and that the kinetics change over the course of experimental differentiation into motor neuron-like cells (Figure 3C). Furthermore, they provide evidence that channels cloned for mES display faster kinetics when heterologously expressed in HEK293 (Figure 6). These observations provide a strong foundation, but the work would have considerably more impact if the authors could provide more mechanistic information about how these two cellular environments alter the kinetics of desensitization.

---

## [Author Response]

Summary:This is an interesting and well executed study showing that mouse embryonic stems cells express mechanosensitive ion channels that exhibit slowly desensitizing properties in response to poking with mechanically actuated probe. Measurements of whole-cell mechanosensitive currents with different cations in the extracellular solution suggest that the channels are relatively non-selective between cations, with a small preference for calcium over sodium or potassium. Single channel measurements with excised patches show similar kinetic properties when elicited with pressure-clamp and compared to whole-cell measurements obtained with the poking assay, and they show that the mechanosensitive channels have a single channel conductance of ~25 pS. When ES cells are differentiated into motor neurons the mechanosensitive currents become more rapidly desensitizing before they disappear, as voltage-gated conductances become larger. Interestingly, the authors show that expression of Piezo1 correlates with the amplitude of the mechanosensitive currents during differentiation, and KO of Piezo1 leads to loss of mechanosensitive currents. Finally, the authors clone Piezo1 from the ES cells and although they identify three residues differing from the previously cloned Piezo1 subunit, expression in mammalian cells gives rise to rapidly desensitizing currents that resemble those previously reported for the Piezo1 channel. The interesting take home message is that a regulatory mechanism must exist in ES cells that converts Piezo1 from rapidly desensitizing to slowly desensitizing. The manuscript was easy to read and comprehend and it makes a valuable contribution to the literature.Although all three reviewers were enthusiastic about this study, the consensus was that the work would be more appropriate for eLife if the authors could provide evidence for what function Piezo1 might have in mES cells or for the mechanism by which extrinsic factors might be shaping Piezo1 kinetics in different cellular environments.

We thank the reviews for their helpful comments on our manuscript. We have addressed all of the points by either modifying the text, adding experiments, or commenting on the issue.

A major point brought up by the reviewers concerned the lack of a functional model that makes sense of the expression of Piezo1 in stem cells. To address this issue, we attempted several lines of experiments aimed to address what the potential role of Piezo1 in stem cells is. Unfortunately, we were unable to find a direct effect of Piezo1 expression in several quantitative and qualitative measures of stem cell differentiation. However, we did note a proliferation defect in Piezo1 knockout cells, indicating that expression of Piezo1 could be involved in setting the normal division rate of cells. We must point out that while our efforts tried to span a broad set of conditions, there is as of yet no clear model for the role of mechanosensory processes in early stem cells, and therefore it is possible that the role of Piezo1 could be involved in untested conditions.

Another major point concerned the lack of model explaining how the Piezo1 channel changes its kinetics in different cellular contexts. We believe that, although of major interest, it escapes the scope of our current work. Below we address the reviewer’s comments in detail.

Essential revisions:1) Piezo1 has known to be widely expressed, so the observation they are in mES is interesting, but perhaps not particularly surprising. More to the point, both Piezo1 and Piezo2 channels have already been shown to have different kinetics depending on the cell types where they are found. Piezo2-dependent currents in Merkel cells have a shoulder current not commonly observed in sensory neurons (Woo et al., 2014; Ikeda et al., 2014). Very recently it was shown that Piezo2 mediates at least some of the slower adapting currents in duck sensory neurons (Schneider et al., 2017). Perhaps more to the point, the currents reported here look remarkably similar to those previously reported to be found in some types of HEK293 cells (Dubin et al., 2017).

Thank you for the comment, we have added the recommended references to broaden our cited sources. Conceptually, indeed, Piezo1 has been shown to be expressed in several cell types, and its kinetics have been shown to be modestly variable. For example, in Schneider et al. (2017), the authors report a shift in inactivation constant from 3.1 ± 0.17 ms in mouse Piezo2 to 5.1 ± 0.47 ms in duck Piezo2. The other reports do not measure inactivation constants but from the figures they appear to be in the same range. Though these reports obviously bear great significance, to our knowledge, no paper has reported such dramatically slowed kinetics – turning the channel at times into a non-inactivating channel – for Piezo1 as we can unequivocally attribute exclusively to Piezo1 in stem cells. In addition, we believe that the discovery of a specific developmental program that drastically modifies the desensitizing behavior of Piezo suggests the relevance of the regulation of Piezo kinetics as an unexplored novel regulatory mechanism. Finally, although the Piezo1 gene has been shown to be widely expressed, as the reviewer rightly points out, the level of current that we found in mES cells is many times greater in magnitude larger than those observed in other non-sensory types such as HEK293 cells. This is interesting, given that mouse embryonic stem cells are the most ‘stem’ state of mammalian cells, that can give rise to any other cell type and whose development program is still widely undescribed. The potential effect of mechanosensory components in the development of embryos is as of yet understudied.

2) While the authors discuss possible implications of Piezo1 function in mES cells (Discussion section), they do not attempt to test those hypotheses experimentally. This is puzzling given the authors took the time to create Piezo1 KO mES cells. There seems to be a missed opportunity to experimentally examine the role of mechanosensitive signaling in mES cells differentiation, outgrowth, survival etc. At a minimum, the authors could have tested their hypothesis about a role for Piezo1 in sensing substrate stiffness.

We appreciate the encouragement to pursue these experiments, and in response have done extensive additional research. The new data has been added as Figure 7 and Table 1, and the resulting conclusions are now incorporated in the relevant sections. As a summary, we compared Piezo1 knockout to wild type mES cells at pluripotency and during early differentiation. Using both qualitative and quantitative measures we found a proliferation defect in the Piezo1 knock out mES cells, but no difference between both phenotypes in any other parameter. We can attribute the proliferation defect to cell cycle interphase lengthening, but we were not able to link it to any mechanosensory process. As suggested by the reviewer, our experiments also included testing mES response to differential substrate stiffness, however Piezo1 genotype did not affect this response to the tested substrates. While we were unable to find a specific role for Piezo1 in mES cells, it is also unclear what the overall role of mechanosensory processes in stem cell differentiation is, and therefore it is possible that we have not tested the appropriate parameters, as it is unknown which assay would be the appropriate one.

3) The authors do a nice job of showing that the mES mechanosensitive currents are mediated by Piezo1 (Figure 5D) and that the kinetics change over the course of experimental differentiation into motor neuron-like cells (Figure 3C). Furthermore, they provide evidence that channels cloned for mES display faster kinetics when heterologously expressed in HEK293 (Figure 6). These observations provide a strong foundation, but the work would have considerably more impact if the authors could provide more mechanistic information about how these two cellular environments alter the kinetics of desensitization.

We kindly thank the reviewers for their overall positive comments. We agree with the observation that a more impactful result would derive from studying the contribution of cellular environment to channel function, however, we think that the initial observation and careful testing reported in this work stands on its own to contribute to the increasingly complex field of mechanosensory ion channels.